

# Diminishing returns as a function of disturbance

Kimmo Sorjonen and Bo Melin

Department of Clinical Neuroscience, Karolinska Institutet, Stockholm, Sweden

## ABSTRACT

According to Spearman's law of diminishing returns (SLODR), IQ test scores are more $g$ saturated among those with low, compared to those with high, ability. The present simulation shows that such difference in saturation can be observed if test scores are affected by a disturbing factor, for example, low motivation, illness, or linguistic confusion, that varies in magnitude between individuals. More contemporary criteria of SLODR can also be satisfied if test scores are affected by disturbance, especially if the disturbance variable is negatively skewed. This indicates a possible threat against the validity of findings supporting SLODR and points at the importance for researchers to try to eliminate the influence of such disturbing factors from their studies.

## INTRODUCTION

To measure abilities can be tricky, as performances can be assumed to be affected not only by peoples' abilities but also by various disturbing factors. That is why athletes' performances tend to fluctuate more than their underlying true abilities can be supposed to do. Here the disturbing factors could be such things as bad weather, a sore knee, or lack of motivation. Performances on tests of cognitive or creative abilities, for instance, could also be influenced by disturbing factors, such as low motivation, illness, or linguistic confusion.

According to the intelligence-creativity threshold hypothesis, intelligence should have a positive influence on creative potential but only up to a certain level, the threshold is often set at IQ 120, after which a further increase in intelligence should have no additional effect on creative potential. This hypothesis has received some empirical support (*Cho et al., 2010*; *Jauk et al., 2013*; *Shi et al., 2017*, but see *Kim, 2005*, for a non-confirming meta-analysis).

However, a simulation by *Sorjonen, Ingre & Melin (2019)* indicates a high probability to observe such threshold-like associations if participants experience varying degrees of disturbance that affects their performances both on the tests of intelligence and on the tests of creativity. A high degree of disturbance will tend to attenuate the association between measured intelligence and creativity among those with high true abilities while it accentuates the association among those with low abilities.

Corresponding author
Kimmo Sorjonen,
kimmo.sorjonen@ki.se

*Spearman (1927)* "law of diminishing returns" (SLODR), also called the "differentiation hypothesis" (*Garrett, 1946*), has often been evaluated, and demonstrated, with a method where the *g* saturation of intelligence test scores, which corresponds to the strength of the correlations between the test scores, has been compared between groups assumed to have high or low average *g*. An often-observed lower degree of *g* saturation in the former group has been taken to indicate that intelligence is more differentiated/specialized among those with high *g*, with some individuals having, for example, especially high verbal or spatial intelligence. A higher *g* saturation among those with assumed low *g*, on the other hand, has been seen to suggest that in this subgroup intelligence is less specialized and more one-dimensional.

For example, *Deary et al. (1996)* claimed support for SLODR after analyzing data from Irish schoolchildren ($N$ = 10,535) on eight Differential Aptitude Tests (DAT) subtests, measuring (1) Verbal reasoning, (2) Numerical ability, (3) Abstract reasoning, (4) Clerical speed and accuracy, (5) Mechanical reasoning, (6) Space relations, (7) Spelling, and (8) Language usage. When dividing the participants into two subgroups, high and low scorers, based on verbal reasoning, the amount of variance in the seven remaining subtests accounted for by a first unrotated principal component equaled 36.2% and 43.6% among the high and low scorers, respectively. When the division was based on numerical ability and space relations the corresponding values were 40.9% and 46.2% for high scorers and 46.2% and 52.9% for low scorers, respectively. Other studies claiming support for SLODR include *Detterman & Daniel (1989)*, *Legree, Pifer & Grafton (1996)*, *Reynolds & Keith (2007)*, *Tucker-Drob (2009)*, as well as a meta-analysis by *Blum & Holling (2017)*.

However, the traditional method of calculating the amount of variance accounted for by a first unrotated principal component, or the average intersubtest correlation, in subgroups assumed to differ in *g* has been criticized. It has, for example, been claimed that this method can give confounded results possibly affected by the arbitrary decision concerning how to form the subgroups or by the skewness of test scores (*Molenaar et al., 2010*; *Murray, Dixon & Johnson, 2013*). According to more contemporary thoughts, ability differentiation, i.e., diminishing returns, might be operationalized as (1) a negatively skewed latent *g*, (2) heteroscedastic subtest residuals, with a larger residual variance among those with high compared to low *g*, and (3) non-linear *g* loadings (*Hessen & Dolan, 2009*; *Molenaar, Dolan & Van der Maas, 2011*; *Molenaar, Dolan & Verhelst, 2010*; *Murray, Dixon & Johnson, 2013*; *Tucker-Drob, 2009*). The fulfillment of these criteria can be analyzed with structural equation modeling (SEM) based methods.

Researchers using such more contemporary methods/operationalizations have questioned if there really is any convincing support for SLODR. For example, when using a traditional method, *Murray, Dixon & Johnson (2013)* found that support for SLODR was proportional to the average skew of the test battery, that is, was mainly found in positively skewed batteries. When using more contemporary methods, they found heteroscedastic residuals, positively correlated with *g*, in positively skewed test batteries, while the criterion of non-linear *g*-loadings, on the contrary, was found in negatively skewed batteries. The criterion of a negatively skewed latent *g* was never fulfilled. In analyses of data from

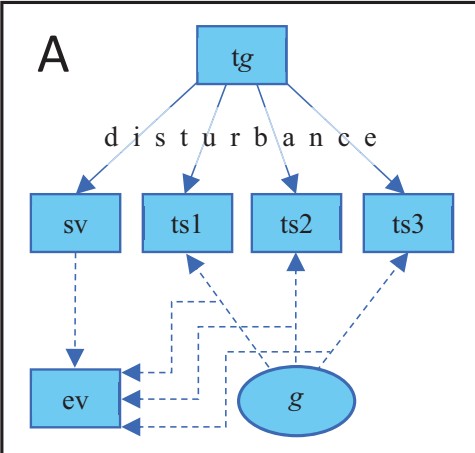
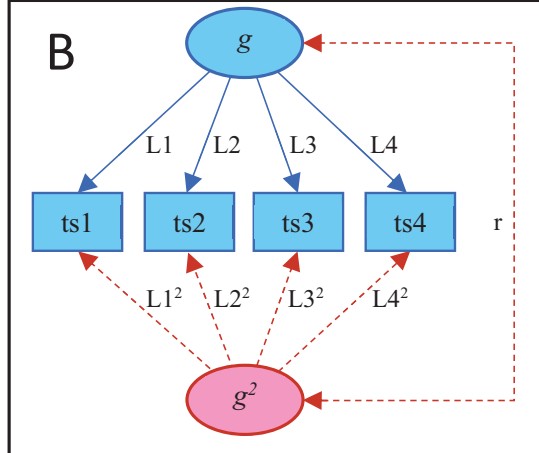

**Figure 1 Steps in the simulation and analyses.** (A) Steps in the simulation (solid lines) and analyzed parameters (dashed lines) of the traditional paradigm. Test scores and a selection variable (sv) are affected by true $g$ (tg) but the effect is attenuated by disturbance that varies in magnitude between individuals. Amount of variance in the test scores explained by the first un-rotated principal component (ev) is calculated separately in the subgroups defined by the selection variable. Note: 14 test scores, besides the selection variable, were used in the actual simulation. (B) Illustration of the analyses when evaluating more contemporary criteria of SLODR. Test scores are regressed on either one single common $g$ factor (blue part) or on one linear and one quadratic $g$ factor (blue + red part). In the latter case, factor loadings for the quadratic factor are constrained to equal the square of the corresponding linear factor loadings. Note: 15 test scores were used in the actual simulation.

young Spanish adults ($N$ = 588) on 14 WAIS-III subtests, *Molenaar, Dolan & Van der Maas (2011)* found some indications of quadratic second-order factor loadings and a negatively skewed second order common factor ($g$). However, contrary to predictions by SLODR, residual variances tended to decrease with increased $g$.

As mentioned above, simulations have indicated a high probability to observe threshold-like associations, for example between intelligence and creativity, if participants experience varying degrees of disturbance that affects their performances (*Sorjonen, Ingre & Melin, 2019*). Due to its similarity to the intelligence-creativity threshold hypothesis, with both proposing a stronger association between performances among those with low ability, it is possible that SLODR might be sensitive to the influence of disturbing factors as well. The objective of the present simulation study was to assess this possibility. Although the traditional method to analyze SLODR through degree of $g$ saturation in subgroups based on different levels of $g$ has been deemed untrustworthy (see above), we still include it in the present study in order to see how it might be affected by disturbance in the measurement of intelligence. We do this due to personal curiosity and because the majority of all studies of SLODR have used this method. They do, for example, seem to constitute most of the studies in *Blum & Holling (2017)* meta-analysis. However, anyone not wishing to know how the traditional method is affected by disturbance should feel free to skip the sections with the heading "Traditional paradigm" below, along with the first paragraph in the discussion, and Figs. 1A and 2.

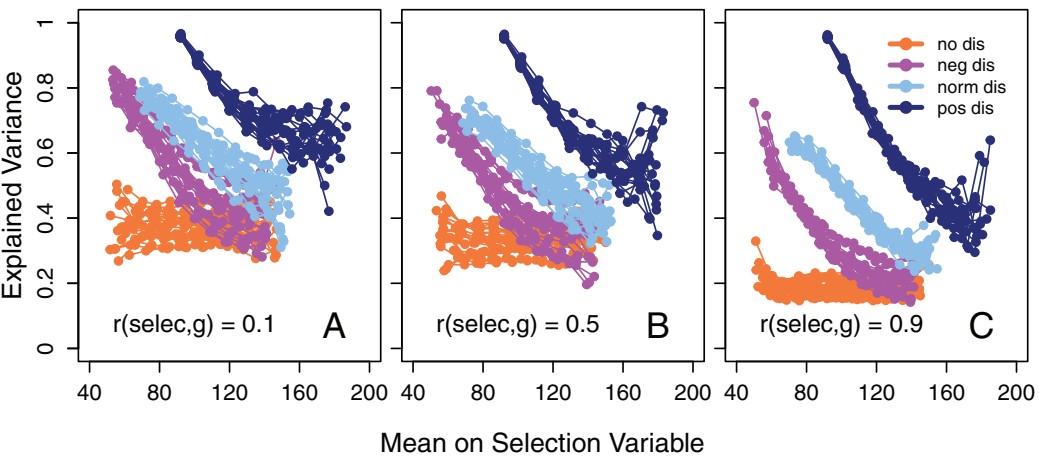

**Figure 2 Explained variance as a function of mean intelligence.** Amount of variance in 14 tests explained by the first un-rotated principal component as a function of the mean value of the selection variable in the subgroups when the disturbance variable is either negatively, normally, or positively distributed or if there is no effect of disturbance on test scores. Separately for three different degrees of correlation between $g$ and the selection variable (before disturbance), namely (A) 0.1, (B) 0.5, and (C) 0.9. The subgroups always have a maximum range of 20 points on the selection variable. 20 simulations (adjoined by a line) for each distribution of the disturbance variable in each part and with $N = 10,000$ in each.                                               

## METHOD

### Traditional paradigm

Using R 3.5.0 statistical software (*R Core Team, 2018*) and the psych package (*Revelle, 2018*), data was simulated and analyzed through the following steps (Fig. 1A, script available at https://osf.io/7ydp8/): (1) 10,000 virtual participants were allocated a true $g$ score from a random normal distribution ($M = 100$, SD = 15); (2) The participants were allocated 15 test scores ($M = 100$, SD = 15) with a defined population correlation with the $g$ score. The correlation for the first test score, the selection variable, was set to either 0.1, 0.5, or 0.9 while the correlations for the other 14 variables were drawn from a random uniform distribution between 0.2 and 0.9. Being continuous, these test scores could be seen to correspond to first order factor scores or subtest scores rather than to traditional dichotomous test items and, consequently, the calculated factor loadings correspond to second order factor loadings; (3) The virtual participants were allocated a disturbance score from a random beta distribution that was either negatively (alpha = 9, beta = 1) or positively (alpha = 1, beta = 9) skewed or approximately normally distributed (alpha = 9, beta = 9). These disturbance scores varied between 0 and 1 with a lower value indicating a higher degree of disturbance; (4) The virtual participants' test scores, including the selection variable, were multiplied by their disturbance score, except in the cases with no effect of disturbance; (5) Based on their value on the selection variable (rescaled to the conventional $M = 100$, SD = 15), the virtual participants were divided into 10 subgroups within 20 points on the selection variable. Hence, there was overlap between the subgroups, with some participants included in more than one group; (6) In each subgroup, the amount of variance in the test scores, excluding the selection variable, that could be

explained by a first un-rotated principal component, that is $g$ saturation, was calculated. According to SLODR, there should be a negative association between the mean of the selection variable and the degree of $g$ saturation in the subgroups.

This procedure was repeated 20 times for each distribution of the disturbance variable and when disturbance had no effect, and for the three levels of correlation between the selection variable (undisturbed) and $g$. When using one of the test scores for creating the groups and then not including this in the principal component analysis, we follow the procedure by *Deary et al. (1996)*.

### More contemporary operationalization

As mentioned in the introduction, a more contemporary operationalization of SLODR include the criteria (1) a negatively skewed latent $g$, (2) heteroscedastic subtest residuals, with a larger residual variance among those with high compared to low $g$, and (3) non-linear $g$ loadings. Data was simulated similarly as described above, bar the selection variable, with one thousand simulations ($N = 10,000$ in each) for each of the four types of disturbance variables—non-existent, normally distributed, negatively, and positively skewed. Data was analyzed using the lavaan (*Rosseel, 2012*) and semTools (*Jorgensen et al., 2018*) packages in R.

Initially, a one-factor confirmatory factor analysis (CFA) was applied to data (the blue part in Fig. 1B). Using the lavPredict function in lavaan, $g$ was estimated for each virtual participant and the skewness, with significance, of this $g$ was calculated with the skew function in semTools. According to SLODR, $g$ should be significantly negatively skewed.

Employing the lavPredict function once more, the predicted score, and then the squared difference between the observed and this predicted score, on each of the fifteen tests was calculated for each virtual participant. The individual mean of these squared differences correspond to the individual residual variance, that is, variance in the test scores that is not accounted for by the individual's $g$. According to SLODR, there should be a significant positive correlation between $g$ and this residual variance. Following *Hessen & Dolan (2009)*, the logarithm of residual variance was used in this analysis.

For the third criterion, the test scores were regressed on a quadratic latent $g$ factor in addition to the linear latent $g$ factor (red part in Fig. 1B). The factor loadings of quadratic $g$ were constrained to be the square of the corresponding linear factor loadings. In this way, the quadratic $g$ factor catches the acceleration in the effect of $g$ on the test scores. According to SLODR, the addition of quadratic $g$ to the model should result in significant improvement of model fit. Moreover, the estimated value on the quadratic $g$ factor should become increasingly more negative with an increase in $g$ (= diminishing returns), i.e. there should be a significant negative correlation between $g$ and $g^2$.

## RESULTS

### Traditional paradigm

In accordance with SLODR, a negative association between the mean value of the selection variable in subgroups and the amount of variance in the test scores that could be explained by a first un-rotated principal component, that is, $g$ saturation, was apparent, except

**Table 1 Fulfillment of contemporary SLODR criteria.** Proportions of a significantly skewed common factor ($g$), correlation between $g$ and residual variance, quadratic factor loadings, correlation between $g$ and the quadratic term ($g^2$), and fulfillment of all crucial criteria as functions of the distribution of the disturbance variable.

| Criteria | Disturbance | | | |
|---|---|---|---|---|
| | No | Negative | Normal | Positive |
| Positive skew | 0.030 | 0.000 | 1.000 | 1.000 |
| **Negative skew** | 0.027 | 1.000 | 0.000 | 0.000 |
| **$r(g, \text{res.})^a > 0$** | 0.021 | 1.000 | 1.000 | 1.000 |
| $r(g, \text{res.})^a < 0$ | 0.032 | 0.000 | 0.000 | 0.000 |
| **$p(g^2)^b < 0.05$** | 0.031 | 0.997 | 0.998 | 0.945 |
| $r(g, g^2)^c > 0$ | 0.010 | 0.000 | 0.000 | 0.000 |
| **$r(g, g^2)^c < 0$** | 0.012 | 0.811 | 0.974 | 0.940 |
| All$^d$ | 0.000 | 0.811 | 0.000 | 0.000 |

**Notes:**
1,000 simulations in each column.
[a] Correlation between $g$ and the logarithm of residual variance.
[b] Significance when adding a quadratic term to the factor model.
[c] Correlation between $g$ and the quadratic term.
[d] Fulfilment of all crucial criteria (in bold).

when disturbance was not allowed to affect the test scores (Fig. 2). When the selection variable (undisturbed) was strongly correlated with the true $g$ score (Fig. 2C) this association was accentuated, but we also see a general decrease in the amount of explained variance compared to when the correlation was weaker (Figs. 2A and 2B). This is expected, because in the former case the participants are more homogeneous in their $g$ and there is less room for a common factor to explain variance in the test scores. With a positively skewed disturbance factor the selection variable also became positively skewed, with few participants at the high end of the scale, and here we see a high degree of instability in the calculated amount of explained variance.

## More contemporary operationalization

In Table 1, we see that with no disturbance affecting test scores, the probability to fulfill the criteria of a negatively skewed $g$ and a positive correlation between $g$ and residual variance are close to what can be expected given a significance level of 0.05. Fulfillment of the criterion of non-linear $g$ loadings, which requires both a significant improvement in model fit when adding a quadratic $g$ factor to the model and a significant negative correlation between $g$ and $g^2$ (see the "Method"), is less probable.

With a negatively skewed disturbance variable affecting test scores, all of the contemporary criteria are fulfilled in 81% of the cases, and without a requirement for a negative correlation between $g$ and $g^2$ we would see virtually perfect fulfillment. With a normally distributed or a positively skewed disturbance variable, the probability for a positive correlation between $g$ and residual variance and for a negative correlation between $g$ and $g^2$ are extremely high. However, here we always see, contrary to SLODR criteria, a positively skewed $g$ (Table 1).

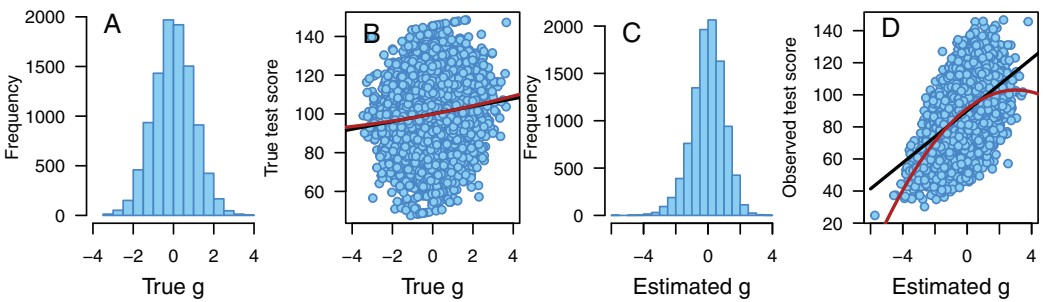

**Figure 3 Illustration of the effect of disturbance.** Illustration of the effect of a negatively skewed disturbance variable. (A) A perfectly normally distributed true $g$ has (B) a weak linear association with the true score on one of the fifteen tests. When test scores are affected by a negatively skewed disturbance variable, we observe (C) a negatively skewed estimated $g$ that has (D) a quadratic association with the observed test score. We also observe heteroscedastic residuals that increase with increased $g$ (D). $N = 10,000$ in each part, black line = best fitting linear association, red line = best fitting quadratic association.               

The effect of a negatively skewed disturbance variable is illustrated in Fig. 3. In this example, true $g$ is almost perfectly normally distributed (skewness = −0.001, $p = 0.982$, Fig. 3A) and has a weak positive linear association with the true score on one of the fifteen tests ($r = 0.131$, $p < 0.001$, Fig. 3B). However, if the test scores are affected by a negatively skewed (skewness = −1.521, $p < 0.001$) disturbance variable, the estimated $g$ becomes negatively skewed (skewness = −0.365, $p < 0.001$, Fig. 3C). Now we also see slightly heteroscedastic residuals that increase with estimated $g$ (correlation between $g$ and the logarithm of squared deviations from the linear prediction line in Fig. 3D = 0.036, $p < 0.001$, note that the correlation between $g$ and the logarithm of individual residual variance over all fifteen test scores is substantially stronger, $r = 0.218$, $p < 0.001$) and a quadratic association between estimated $g$ and observed test score ($B = −1.266$ and $p < 0.001$ for the quadratic regression term, Fig. 3D).

## DISCUSSION

The present simulation shows that if test scores are affected by disturbance—which could be, for example, low motivation, illness, or linguistic confusion—that vary in magnitude between individuals, we will, similarly to studies using a traditional paradigm to assess SLODR, tend to see a higher degree of factor saturation among those with a low score on an associated selection variable compared to those with a high score. This is true irrespective of the distribution of the disturbance variable. Consequently, the present findings could be added to the growing number of warnings that a positive result when using the traditional paradigm cannot be trusted to indicate a true SLODR-effect.

When evaluating the effect of disturbance using more contemporary criteria, results in accordance with SLODR were seen mainly with a negatively skewed disturbance variable. With a normally distributed or a positively skewed disturbance variable, two of three criteria, namely heteroscedastic residual variance and quadratic factor loadings, were usually fulfilled, while the skewness of estimated $g$ was in the opposite direction compared

to SLODR criteria. Hence, one of the advantages of these contemporary criteria seems to be that one of them, negative skewness of estimated *g*, offers protection against type 1-errors, unless test scores have been affected by negatively skewed disturbance, while the traditional method seems to deliver false support for SLODR irrespective of the distribution of the disturbance variable. However, negatively skewed disturbance would mean that most participants receive test scores close to their true values while a few perform well below their true ability, and we believe that such lack of validity among a minority of participants can be quite realistic in many test situations.

According to *Murray, Dixon & Johnson (2013)*, the empirical support for SLODR can be called into question as it is based mainly on studies employing traditional paradigms, which cannot discriminate between a true SLODR-effect and effects of subtest characteristics, for example, skewness. The present simulation throws some additional pessimism into the pot. If using more contemporary methods and finding support for SLODR in the form of a negatively skewed latent *g*, heteroscedastic residuals, and quadratic factor loadings, one cannot be sure if this is due to a true SLODR-effect or the influence of some negatively skewed disturbance factor that varies in degree between study participants and that affects their test scores.

We do not, of course, claim to have proven that all findings in accordance with SLODR are due to the influence of negatively skewed disturbance. We merely point at the possibility that this could have given SLODR a helping hand in some studies. This points at the importance for researchers to try to eliminate, as far as possible, the influence of all such disturbing factors from their studies. However, we can probably never be absolutely sure that we are measuring all of the participants' true abilities rather than merely their performances on the day.

Still, even if findings that support SLODR would be due to the influence of disturbance, this does not necessarily mean that SLODR is just an artifact. For example, according to *Detterman (1993)* explanation of SLODR, as well as the more recent process-overlap theory (POT, *Kovacs & Conway, 2016*, *2019*), the "disturbing factor" could be a deficit in some important general cognitive process which restrains the functioning of the whole system. Without this deficit, test scores are mainly due to more specific processes and, consequently, less correlated and *g* saturated. If Detterman's hypothesis and POT are correct, and if studies using more contemporary criteria start supporting SLODR (it does not look very promising so far), one of the contributions of the present simulation would be to predict that most people experience only low degrees of this deficit while a few experiences a very high level.

## CONCLUSIONS

The present simulation indicates that findings that falsely seem to support Spearman's law of diminishing returns (SLODR) can be obtained if test scores are affected by a disturbance variable that varies in magnitude between respondents. This is true for contemporary operationalizations of SLODR and even more so if using a traditional paradigm to analyze data.

### Funding

The authors received no funding for this work.

### Competing Interests

The authors declare that they have no competing interests.

### Author Contributions

- Kimmo Sorjonen conceived and designed the experiments, performed the experiments, analyzed the data, prepared figures and/or tables, authored or reviewed drafts of the paper, and approved the final draft.
- Bo Melin conceived and designed the experiments, authored or reviewed drafts of the paper, and approved the final draft.

### Data Availability

Script (which also generates the data) is available at OSF: Sorjonen, Kimmo. 2020. "Diminishing Returns as a Function of Disturbance." OSF. May 23. osf.io/7ydp8.

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
