# Peer review of "Diminishing returns as a function of disturbance"

_PeerJ, doi:10.7717/peerj.9490_

## Round 0.1 · original submission · Major Revisions

Three reviewers have provided comments on your work. I like the brevity of your manuscript, although I think that responding to some of their comments would lead to a, at least slightly, longer one, and I appreciated the appropriate caution in your interpretation of the results. As you will see, the reviewers have presented a wide range of comments. Reviewer #1 makes some favourable comments about your manuscript but importantly raises some challenging methods questions. You will need to either justify your chosen approach or change your approach to one of those they suggest (or another, perhaps of your own devising). Related to this, they also identified gaps in the literature review that would need to be addressed either way. Reviewer #2 is more positive and makes some suggestions around the literature. As they note, they are an author of some of the articles they suggest and you should make your own decisions around whether each can help tell your particular story (for what it’s worth, I think at least some of these could be very useful additions). I liked their suggestions around interpreting “disturbance” and hope that you will also find these useful. Reviewer #3 makes a number of useful suggestions, many of which should make the manuscript easier to follow for a wider range of readers, each of which warrants careful consideration. Their comments about the methods seemed important to me and their suggestions around reporting should help the reader to follow your results. Overall, I think that all of the comments from these three reviewers deserve careful consideration and response, either through making and explaining changes to your manuscript or by thoughtful rebuttal.

There are a few typos (e.g. “for SLODR” on Line 141 but “for the SLODR” on Line 145; and “This fits with the results in an earlier simulation, w[h]ere we found…” on Lines 152–153) but nothing that some careful proofreading wouldn’t catch and correct. For Figure 4, can you show the inequality sign as a single character (“≥”) rather than two separate characters (“>=”)? (You could try one of ‘expression("">=0.05)’ or ‘"\u2265 0.05"’ for this.)

Reviewer 1 ·

Basic reporting

This manuscript is on the question whether skewed residuals can cause SLODR to arise in intelligence datasets. Generally, the manuscript is well written and understandable. I liked reading this manuscript.

Experimental design

The methods used in this study are outdated. That is, the authors used principal component analysis (PCA) following (Deary et al, 1996). There are many more recent paper that present well-established and well-accepted methods to study differentiation. E.g.:

The mixture approach by Reynolds et al
Reynolds, M. R., Keith, T. Z., & Beretvas, N. (2010). Use of factor mixture modeling to capture Spearman's law of diminishing returns. Intelligence, 38, 231−241.

The non-linear factor analysis approach:
Tucker-Drob, E. M. (2009). Differentiation of cognitive abilities across the life span. Developmental Psychology, 45, 1097−1118.
Molenaar, D., Kő, N., Rózsa, S., & Mészáros, A. (2017). Differentiation of cognitive abilities in the WAIS-IV at the item level. Intelligence, 65, 48-59

The moderated factor model approach:
Molenaar, D., Dolan, C.V., Wicherts, J.M., & van der Maas, H.L.J. (2010). Modeling Differentiation of Cognitive Abilities within the Higher-Order Factor Model using Moderated Factor Analysis. Intelligence, 38, 611-624.

Most importantly, it is well established, that PCA is a suboptimal method to study intelligence in general (see e.g., Dolan, 2000) and SLODR in particular (see e.g., Molenaar et al 2010 above). Key of the problem is that PCA does not explicitly separate the residuals from the g-variance and does not explicitly model the intelligence structure

Dolan, C. V. (2000). Investigating Spearman's hypothesis by means of multi-group confirmatory factor analysis. Multivariate Behavioral Research, 35(1), 21-50.

Validity of the findings

Given the above, the results cannot be considered valid. As for instance demonstrated by Molenaar et al (2010), PCA can give very misleading results concerning SLODR. Ideally, the present research question should be studied in one of the contemporary formal modeling approaches above for the results to be interpretable. (i.e., if data is simulated according to one of the models above and the residuals are skewed, how does this skewness show up in the model?) or ofcourse in a new and appropriate formal model.

Additional comments

The literature review is not very thorough. Many studied have been done into SLODR and spurious results (e.g., Murray, Dixon, & Johnson, 2013; Tucker-Dob, 2009, Molenaar et al., 2017), and many advanced modeling techniques have been developed to counter these spurious effects.

·

Basic reporting

This is an interesting paper testing a fascinating idea: that ability differentiation is the result of a disturbing factor which is possibly negatively skewed. (SLODR and differentiation are synonymously used in the literature, I prefer differentiation because SLODR with its “diminishing returns” analogy from economics implies that one “invests in” one’s g which is already interpretative.)

It is an embarrassing moment for me because as an author I am not greatly fond of reviewers who use the opportunity to shamelessly promote their own work. However, in this particular instance, I cannot help but point to papers which I authored and which are, I believe, greatly relevant both empirically and theoretically.

The authors point out that a kind of disturbance could be a deficiency in some central system. This is most pronounced at the end of the paper (162-170):

“Still, even if findings that support SLODR would be due to the influence of disturbance, this does not necessarily mean that SLODR is just an artefact. For example, according to Detterman's (1993) explanation of SLODR, the “disturbing factor” could be a deficit in some important general cognitive process which restrains the functioning of the whole system. Without this deficit, test scores are mainly due to more specific processes and, consequently, less correlated and g saturated. If Detterman’s hypothesis is correct, and if studies using more contemporary criteria start supporting SLODR (it does not look very promising so far), one of the contributions of the present simulation would be to predict that most people experience only
low degrees of this deficit while a few experiences a very high level.”

Detterman’s explanation is actually similar to process overlap theory, a more recent account of the positive manifold and differentiation. See:

Kovacs, K., & Conway, A. R. A. (2016). Process overlap theory: A unified account of the general factor of intelligence. Psychological Inquiry, 27, 151–177.

Kovacs, K. & Conway, A. R. A. (2019). What is IQ? Life beyond general intelligence. Current Directions in Psychological Science, 28, 189-194.

Please allow me to quote from the latter paper:

“This explanation, called process-overlap theory
(POT), does not propose a general cognitive ability and
therefore does not endorse psychological g. Instead, it
proposes that intelligence is determined by multiple
components, both domain-general and domain-specific.
Certain domain-general processes (often called executive
functions in the literature, e.g., Diamond, 2012)
overlap with domain-specific processes during mental
test performance. These executive processes are central
to human intelligence in the sense that they are tapped
by a large number of tests. Specific processes, on the
other hand, are mostly tapped by tests with corresponding
specific (verbal, spatial, etc.) content only.
Executive processes thus function as a bottleneck:
They constrain performance in a wide variety of tests
that tap different domains. Hence, performance in these
different, specific domains (verbal, spatial, etc.) correlate,
and the positive manifold emerges. Moreover,
the lower the level of executive functions, the stronger
the bottleneck effect and the higher the correlations
between diverse tests: This, according to POT, is the
explanation of ability differentiation.”

and:

“This executive bottleneck, represented in a formal
mathematical model, is the primary proposal of POT
(Kovacs & Conway, 2016). Besides the positive manifold
itself, it also explains ability differentiation and the
worst-performance rule. People who generally perform
below average on cognitive tasks (low ability) are likely
to exhibit deficits in executive processes that impact
performance across a range of domains. In contrast,
people who generally perform above average on cognitive
tasks (high ability) are unlikely to exhibit deficits
in executive processes, so their performance primarily
reflects domain-specific processes. That is, the lower
the level of executive functions, the stronger the bottleneck
effect and the higher the correlations between
diverse tests: This, according to POT, is the explanation
of ability differentiation.”

As the latter quote says, this bottleneck effect is formalised a mathematical model in the Psych Inquiry paper. It is in fact a multi-dimensional item response theory model which specifies the probability of correctly answering a test item as the function of passing two independent dimensions. Under this account, it is the domain-general dimension, limiting performance in different domains, that is equivalent of a “disturbance” in the MS.

The Current Directions paper just came out very recently, but the Psych Inquiry paper appeared in more than 3 years ago and, according to Google Scholar, has already been cited 126 times, so I like to believe that it is not due to my horrible narcissism that I think it should be mentioned.

Moreover, it was a strong prediction of the theory that differentiation must manifest itself in working memory capacity, too, and we recently published a paper demonstrating just that on two data sets:

Kovacs, K., Molenaar, D., & Conway, A. R. A. (2019). The domain specificity of working memory is a matter of ability. Journal of Memory and Language, 109, 104048.

Also, differentiation might occur in higher-order models, too, as a decrease of the 2nd order factor loadings as the function of g. This should be discussed.

Kristof Kovacs

Experimental design

Given how accurately it is described in the papers by Molenaar and Tucker-Drob (which are cited in the paper) that the methods described in the MS as the “Traditional paradigm” are suboptimal I see no reason to include them. The sections under “More contemporary operationalisation” already drive the point home, there is nothing the reader gains from additional evidence obtained with methods that are suboptimal, as is acknowledged in the paper.

Validity of the findings

The findings are important for the explanation of differentiation but their theoretical interpretation is not elaborate enough (see more under "Basic reporting").

Also, the authors repeatedly mention three possible sources of disturbance: low motivation, illness, linguistic confusion. Each of these is problematic. For motivation, see

Gignac, G. E. (2018). A moderate financial incentive can increase effort, but not intelligence test performance in adult volunteers. British Journal of Psychology, 109(3), 500-516.

Linguistic confusion as a source of differentiation does not sound likely, as such results were often obtained on test standardisation samples where great care is taken to avoid such confusion. Differentiation in working memory capacity is also hard to explain this way. Illness is a plausible account but without further specification it is somewhat vague. Probably the explanation by Detterman’s system theory (and, if added, process overlap theory) should be discussed other than an endnote.

Overall, these come across as speculative examples that the authors wanted to include for "disturbance" not to look vague.

·

Basic reporting

The manuscript needs more context and elaboration. For example, the introduction should include more information about (a) what “g saturation” means (line 47); (b) how they original hypothesis was operationalized (lines 45-48); (c) why it is still important even though empirical support is lacking (lines 50-51); (d) how the hypothesis has been operationalized more recently, and why it was updated (move lines 84 to 86 to the introduction); and (d) why it is important to compare both the traditional and contemporary operationalizations. Figure 1 is not referenced in the text. Would the figure be different for traditional vs. contemporary hypotheses?

Experimental design

In the methods section there should be more information about (a) why the number of repetitions and/or N varies between different analyses (simulations: line 77 – 20, line 93 – 1000; N: 10,000 for most, N = 500 in Figure 3); and (b) why transformations of the data were needed. Specifically, (1) what was factor analyzed? (line 95); (2) why use the un-rotated and not the rotated principle component? (lines 75-76); (3) why use the logarithm of residual variance and not simply residual variance? (line 97); and (4) why use both g and g-squared in the regression analysis? (line 99). Also, the difference between traditional and contemporary operationalizations appears to involve how the subgroups are formed (lines 72-74 and 84-86) but this is not clear. Finally, I see the contemporary operationalization of the contemporary hypothesis (lines 88-89) but I did not see the equivalent for the traditional hypothesis.

Validity of the findings

I have no doubt about the validity of the findings, as the simulations seem rigorous and reproducible. However, the results should be reported so that the reader knows what pattern(s) to expect (look for) in the tables and figures. What pattern of data tells us whether (a) adding disturbance produced the same results as expected by SLODR (i.e., what patterns are we looking for?); and (b) whether the disturbance model fits the traditional vs. contemporary hypotheses equally well. There is some explanation regarding the contemporary operationalization (lines 116-128). Still, what is “subtest skewness”? What does a pattern of diminishing returns look like? How do we know whether there is “larger residual variance among those with high compared to low g?” And, what is non-linear g loading onto? Regarding the figures, the diagram in Figure 1 should be enlarged. The other figures are beautiful.

Additional comments

My questions and suggestions are meant to help the research reach a broader audience. I appreciate that in the last paragraph the author(s) reflected on the possibility that the disturbance factor “could be a deficit in some important general cognitive process which restrains the functioning of the whole system” (lines 164-165). And, I agree with the statement that regarding the contribution of the current simulations (lines 169-170).

---

## Round 0.2 · Major Revisions

Thank you for your patience during this challenging time.

The original three reviewers have provided comments on your manuscript and, again, there is a range of views presented by them. I still feel that you have the basis of a useful and interesting manuscript, but I don’t think you have yet established and communicated to the reader a clear and compelling motivation for your study or that readers would fully appreciate the implications of your findings as they relate to that motivation.

As I said previously, while I admire brief manuscripts, I think you need to expand parts of your work to explain things for the reader and to respond to the reviewers’ comments, which will be shared by at least some readers. Reviewer #3 makes a good point in noting that their requests for information were not just for their benefit. I think it is worth noting that your edits to the manuscript seem appreciably lighter than your responses to the reviewers’ comments. If you are able to fully respond to the comments below without extending the manuscript, that is of course fine, but I would be surprised if that was the case.

In particular, I don’t think that you have fully addressed Reviewer #1’s comments in your response and will ask you to again consider their original comments alongside their additional comments should you decide to resubmit a revised version of your manuscript (which I hope you will do). If part of the motivation of looking at some methods is to help readers in interpreting the historical literature, that could be useful but this would need to be made very clear and justified to the reader. I was expecting that this would be a significant part of your conclusions, but the abstract ends by noting the importance of researchers eliminating the influence of disturbing factors, which would also be a useful finding but did not seem to match the approach here (for that aim, I think Reviewer #1’s comment about studying methods that are in current use is entirely valid).

While I appreciated your window washing analogy, this point doesn’t seem to be emphasised in the manuscript itself. Doing this would, I think, require the more extensive discussion of both the historical/traditional and contemporary analysis strategies requested by Reviewer #1. Showing how the results vary over a range of analysis approaches (with a range of sophistication) would seem potentially useful to me in reconciling the literature (which includes studies using sub-optimal approaches) as well as motivating the use of the more sophisticated options in the future. If that approach was used, then I think the results from the less sophisticated approaches become much more interesting.

As Reviewer #1 notes, the data you have simulated is ad-hoc and lacks a formal and well-described data generating mechanism. Their suggestions include using an IRT model to provide a basis for the simulations, which seems the most appealing option to me, but other clear and deliberate data generating models could certainly be used and you could, if you wanted to, argue in favour of your current method.

Reviewer #2 echoes part of Reviewer #1’s comments, namely that the role of the traditional/historical approaches needs to be made clear to the reader.

In summary, there are aspects to this manuscript that I find interesting, but there are apparent inconsistencies for me between the study’s objectives, methods, findings, and recommendations. As noted above, and previously, I would imagine that a manuscript that addresses the reviewers’ points would need to be longer in order to establish a clear story from a well-defined problem motivated by the literature through to the manuscript’s findings and their implications. If you can address all of the reviewers’ points, and I believe that there are multiple options available to you in doing so, I think your manuscript would make a useful addition to the literature. I hope to see a revised version of your manuscript in due course that does this.

Reviewer 1 ·

Basic reporting

As I mentioned in my previous review, the literature review is not complete. In the revised version, the authors didn't act upon this comment. As a reply they mention that they refer to the articles I mentioned. Indeed, these articles are referred to but the paper lacks a thorough discussion of these papers (and others). The single page introduction section does not do a sufficient job in explaining the background and highlighting what problem the authors are solving (and why this problem needs to be solved)

Experimental design

The methods are fundamentally flawed. The authors use an ad-hoc approach to simulate their data (they draw g scores from a normal distribution, and next, they draw test scores that are correlated to those g scores and add disturbances).
To study a topic like this, an explicit statistic model is needed from which data is simulated. This can be a factor model or an item response theory model, but not an intuitive, ad-hoc approach like this.

In my review I also pointed out that the authors should use more contemporary approaches to study differentiation. The authors argue that they *do* apply these approaches. This is not true. Although they indeed use the conceptual idea of heteroscedastic residuals and non-linear g loadings, they test for these ideas using -again- ad hoc approaches (e.g. correlating the g scores to the log-residual variances). This is nowadays really not admissible. If you want to test for heteroscedastic residuals you really have to apply a statistical model (e.g., that of Hessen & Dolan), consult model fit, consult standard errors, etc.

Finally, the authors argue in their reply that PCA should be studied as it has been used previously. I dont agree. I dont think we should study what is wrong with the methods we used in the past. We should study what is wrong with the methods we use *currently* and PCA is not one of them.

Validity of the findings

Because of the above, the results of the present paper are not interpretable. The authors should use proper methods and not the ad-hoc procedures that they are currently using.

Additional comments

This paper really lacks rigor both in the theoretical underpinning and in the statistical treatment of the research problem.

·

Basic reporting

The issued raised in the review have been adequately answered. If the traditional methods stay, it should be highlighted that they are suboptimal and only serve the purpose the authors described in the rebuttal: "By including the method and showing that it is very (more so than more contemporary methods) sensitive to the influence of disturbance, we might contribute to putting the traditional method to a well-deserved rest."

Experimental design

Answer is accepted.

Validity of the findings

Answer is accepted.

·

Basic reporting

No comment.

Experimental design

No comment.

Validity of the findings

No comment.

Additional comments

The authors chose to argue away most of our concerns. To be honest, I wasn't asking for the extra information for my sake. Rather, I was expecting them to explain why they chose not to include it, if that was their decision. In any case, I appreciated the inclusion of some of our suggestions.

---

## Round 0.3 · accepted · Accept

Thank you for your very thoughtful revisions and responses. The one remaining reviewer had no further comments on your manuscript.

On my part, I felt that the story was clearer in this version and that readers will come away from reading it with answers/insights regarding the particular questions they personally are most interested in (including about interpreting studies using traditional and/or contemporary methods, and/or about designing their own studies). I feel that the traditional and contemporary results both address likely questions from potential readers of your work in this way and so I will decline your extremely constructive offer to remove the material on traditional approaches.

The query around the data simulation approach, initially raised by a reviewer, is an interesting one, and one that, as a biostatistician, I have some sympathies with. However, I also take your point that your approach here is the same as used elsewhere, including in your recent article, and one which I sometimes use myself. I don’t mean “ad hoc” as a pejorative in this context, nor do I think that the reviewer intended this interpretation. The model you use is well described, in text and code, as you say, so a reader could replicate your experiments. My thinking about the “best” approach to take with data generating models, whatever “best” might mean in this context, is certainly not immutable and while this aspect wasn’t addressed directly in the revised manuscript, I especially take your point about the previous manuscript, which is particularly salient as I can imagine researchers reading them one after the other.

I have noted a small number of edits that are straightforward enough that I’m happy leaving them for you to address in the proofing process as you see fit. In the below, the suggested edits or additions are shown in upper case.

Congratulations in writing another interesting manuscript and one that should lead to some interesting discussion.

Line 41: The word “subjects” is often avoided these days, in favour of “participants” (which I’d suggest using here), “patients”, or “respondents” as appropriate. Since in this case, the “subjects” are virtual and not real, the risk of any actual offence is effectively avoided, but I feel that it is unusual to see this word these days and you might prefer to replace it. See also Lines 91, 110, 111, 117, 121, 124, 125, 148, 152, 175, 177, and 223.
Line 67: “However, the traditional method of calculating THE amount of variance accounted for…”
Line 70: “…possibly affected by the arbitrary decision CONCERNING how to form…” (or OF instead)
Line 72: I suggest expanding “a.k.a.” here in full.
Line 82: “…i.e. WAS mainly found in positively skewed batteries.”
Line 84: “criterium” should be “criterion”. See the same point for Lines 85, 157, and 183.
Line 86: “analyzes” (a verb) should be “analyses” (a noun).
Line 100: “…THE majority of all studies of SLODR have used this method.”
Lines 101–104: While I appreciate the addition of this signpost for readers, I’ll suggest an alternative, less directive, wording of “However, anyone not WISHING to know how the traditional method is affected by disturbance SHOULD FEEL FREE to SKIP the sections with the heading ‘Traditional paradigm’ below, ALONG WITH the first paragraph in the discussion AND panel A in Figure 1 and Figure 2.” for you to consider. This is merely a suggestion and you are welcome to retain the present version if you prefer it, or to use some variant of the two.
Line 129: “…between the mean OF the selection variable…”
Line 144: Note that you don’t have the version number for lavaan later in the reference (Lines 313–314) as you do for semTools (Line 282) and psych (Lines 308–309). Note also that the way the version is provided differs for semTools (Line 282, before the URL) and psych (Lines 308–309, embedded in the URL).
Line 152: “…mean OF these squared differences…”
Line 168: “…the mean value OF the…”
Line 194: “…true g is ALMOST perfectly normally distributed (skewness = -0.001, p = 0.982, panel A)…”
Line 208: “…similarly TO studies using…”
Line 239: Simply as a suggestion, but would “…rather than merely their performances ON THE DAY.” help remind the reader here that these disturbances can/will vary over time?

References: Do you wish to italicise the titles for packages on CRAN, e.g. Lines 281–282 and 307? The citations from “citation("semTools")” and “citation("psych")” suggest this would be the case:
@Manual{,
title = {\texttt{semTools}: {U}seful tools for structural equation modeling},
author = {Terrence D. Jorgensen and Sunthud Pornprasertmanit and Alexander M. Schoemann and Yves Rosseel},
year = {2020},
note = {R package version 0.5-3},
url = {https://CRAN.R-project.org/package=semTools},
}
@Manual{,
title = {psych: Procedures for Psychological, Psychometric, and Personality Research},
author = {William Revelle},
organization = { Northwestern University},
address = { Evanston, Illinois},
year = {2019},
note = {R package version 1.9.12},
url = {https://CRAN.R-project.org/package=psych},
}

Figure 2: “…the mean value OF the…”

·

Basic reporting

no comment

Experimental design

no comment

Validity of the findings

no comment

Additional comments

In the most recent revision, reviewer suggestions were incorporated and as a result the manuscript is more balanced and clearer than previous versions.